# Metabolic Profile of C-Prenyl Coumarins Using Mass Spectrometry-Based Metabolomics

**DOI:** 10.3390/molecules26216558

**Published:** 2021-10-29

**Authors:** Yan Cheng, Xiaofang Ma, Qi Zhao, Chunyan Wang, Dongmei Yan, Fei Li

**Affiliations:** 1Laboratory of Metabolomics and Drug-Induced Liver Injury, Frontiers Science Center for Disease-Related Molecular Network, West China Hospital, Sichuan University, Chengdu 610041, China; Yancheng2020@wchscu.cn (Y.C.); sdauzhaoq@163.com (Q.Z.); kriswcy@163.com (C.W.); 2Academician Workstation, Jiangxi University of Chinese Medicine, Nanchang 330004, China; mxf18942350993@126.com

**Keywords:** C-prenyl coumarins, metabolomics, anti-inflammatory activity, UPLC-ESI-QTOF-MS

## Abstract

C-prenyl coumarins (C-PYCs) are compounds with similar structures and various bioactivities, which are widely distributed in medicinal plants. Until now, the metabolic characterizations of C-PYCs and the relationship between metabolism and bioactivities remain unclear. In this study, ultra-performance chromatography electrospray ionization quadrupole time-of-flight mass spectrometry-based metabolomics (UPLC-ESI-QTOF-MS) was firstly used to determine the metabolic characterizations of three C-PYCs, including meranzin hydrate (MH), isomeranzin (ISM), and meranzin (MER). In total, 52 metabolites were identified, and all of them were found to be novel metabolites. Among these metabolites, 10 were from MH, 22 were from ISM, and 20 were from MER. The major metabolic pathways of these C-PYCs were hydroxylation, dehydrogenation, demethylation, and conjugation with cysteine, *N*-acetylcysteine, and glucuronide. The metabolic rate of MH was much lower than ISM and MER, which was only 27.1% in MLM and 8.7% in HLM, respectively. Additionally, recombinant cytochrome P450 (CYP) screening showed that CYP1A1, 2B6, 3A4, and 3A5 were the major metabolic enzymes involved in the formation of metabolites. Further bioactivity assays indicated that all of these three C-PYCs exhibited anti-inflammatory activity, but the effects of ISM and MER were slightly higher than MH, accompanied by a significant decrease in inflammatory cytokines transcription induced by lipopolysaccharide (LPS) in macrophages RAW 264.7. Taken together, the metabolic characterizations of the three C-PYCs suggested that the side chain of the prenyl group may impact the metabolism and biological activity of C-PYCs.

## 1. Introduction

C-prenylcoumarins (C-PYCs) are a group of coumarins commonly found in medicinal plants, such as Fructus aurantii. Fructus aurantii is a traditional Chinese medicine that has been used for the treatment of indigestion [1,2], obesity [3], hypertension [4], stones [5], and depression [6]. Meranzin hydrate (MH), isomeranzin (ISM), and meranzin (MER) found in Fructus aurantii belong to C-PYCs. The chemical structures of MH, ISM, and MER are similar, and the substituent groups were identified as hydroxyl, carbonyl, and cyclic ether at the position of C2′, respectively (Figure 1A). In terms of bioactivities, MH plays a role in regulating the gastrointestinal tract [7], anti-atherosclerosis [8], and anti-depression [9,10]. ISM and MER exhibit anti-inflammatory [11], anti-mycobacterial [12], and anti-proliferation effects [13]. So far, the metabolic characterizations of these C-PYCs remain unclear.

UPLC-ESI-QTOFMS-based metabolomics is a powerful tool to quickly profile the metabolites of drugs and xenobiotics. The metabolic map of some natural compounds has been determined using this approach [14,15,16]. In this study, the metabolomics approach was used to investigate the metabolic pathways of MH, ISM, and MER. According to their metabolites and the involved drug-metabolizing enzymes, the metabolic characterizations of these three C-PYCs were finally determined. Moreover, the relationship between the metabolism and biological activity of C-PYCs was elucidated.

## 2. Results

### 2.1. In Vitro Metabolism of MH, ISM, and MER by HLM and MLM

Liver microsomes contain a wide variety of drug-metabolizing enzymes and are commonly applied in in vitro metabolism studies. In our study, the three C-PYCs were incubated with mouse liver microsomal (MLM) and human liver microsomal (HLM), and the in vitro metabolites were analyzed by metabolomics. MH, ISM, and MER treatment groups were well separated from the control group in the OPLS-DA score plot. The metabolites were screened through the *S*-plot (Appendix A). Overall, for the in vitro metabolism of MH, two metabolites (Mh2a and Mh3) were detected in HLM, and five metabolites (Mh1, Mh2, Mh2a, Mh3, and Mh4) were found in MLM (Figure 2A,B). For the in vitro metabolism of ISM, 17 metabolites (Mi1, Mi2-Mi2a, Mi3-Mi3a, Mi4-Mi4b, Mi5-Mi5c, Mi7-Mi7c, and Mi8) were detected, of which Mi3, Mi4b, Mi6b, and Mi6c were unique metabolites in MLM, and Mi4, Mi7, Mi7a, and Mi8 were only detected in HLM (Figure 2C,D). Moreover, 11 metabolites (Me1, Me2-Me2b, Me5-Me5b, Me6-Me6a, and Me8) were detected in the in vitro metabolism of MER (Figure 2E,F). Hydroxylation was clearly the major metabolic pathway of C-PYCs in vitro, including hydroxylation–dehydrogenation, hydroxylation, dihydroxylation, and dihydroxylation-hydrogenation.

The metabolic rates of MH, ISM, and MER in HLM were 8.7%, 84.1%, and 24.8%, and they were 27.1%, 81.1%, and 52.6% in MLM, respectively (Figure 3A). The metabolic pathways included hydroxylation, dehydrogenation, and demethylation, and hydroxylation was the major metabolic reaction of C-PYCs. However, only one hydroxylation metabolite was detected in the in vitro metabolism of MH. The number of metabolic pathways of MH, ISM, and MER in in vitro metabolism was summarized (Figure 3B–E).

### 2.2. Metabolomic Profiling of MH, ISM, and MER Metabolites in Mice

UPLC-ESI-QTOF-MS combined with OPLS-DA and S-plot was used to determine the differences between the PYCs treatment groups and the control group. These treated groups were well separated from the control group in the OPLS-DA score plot. The metabolites were screened through the *S*-plot (Figure 1B–D). Overall, seven metabolites were detected in the in vivo metabolism of MH (Appendix A). Metabolites (Mh2, Mh3, Mh5, Mh5a, Mh6, Mh7, and Mh8) were found mainly in urine (Figure 4A). Only metabolite Mh8 was detected in plasma, and none of them were detected in feces. The major metabolic pathways were demethylation, hydroxylation, dehydrogenation, and dehydration. A total of 10 metabolites (Mi1, Mi2a, Mi5, Mi5b-Mi5c, Mi6, Mi9, Mi9a, Mi10, and Mi10a) were detected in the in vivo metabolism of ISM, which were mainly distributed in urine and feces, and Mi1, Mi3, Mi5, Mi5c, Mi9, Mi9a, Mi10, and Mi10a were the major metabolites (Figure 4B). Hydroxylation, dehydrogenation, hydrolysis, demethylation, cysteine, and *N*-acetylcysteine conjugation were the major metabolic pathways (Appendix A). There were 14 metabolites discovered in the in vivo metabolism of MER, mainly in urine and feces. Among them, Me1, Me2b, Me5, Me5b-Me5c, Me10, Me11, and Me12 were the major metabolites (Figure 4C). Additionally, hydroxylation was the major metabolic pathway for the phase I metabolites. Mi6, Me6, and Me6a were hydrolysis products, which were detected in the in vivo metabolism of ISM and MER. Metabolites Mi9, Mi9a, Mi10, Mi10a, Me9, Me9a, Me10, Me10a, Me11, and Me12 were phase II conjugates of C-PYCs, of which cysteine and *N*-acetylcysteine conjugates were common metabolites, while glucuronide and hydroxylation–glucuronide conjugates were the characteristic metabolites of MER (Table 1).

### 2.3. Structure Elucidation of MH, ISM, and MER Metabolites

#### 2.3.1. Structure Elucidation of MH Metabolites

The parent compound Mh0 was eluted at 6.32 min, which was calculated as C_15_H_18_O_5_ based on the accurate mass [M + H]^+^ at *m*/*z* 279.1227^+^ and the major fragmentation ions of *m*/*z* 261^+^, 243^+^, 189^+^, and 175^+^ described in Figure 5A. Mh1 was calculated as C_14_H_16_O_5_ based on the accurate masses [M + H]^+^ at *m*/*z* 265.1070^+^, and it was 14 Da (CH_2_) less than Mh0. The ions of *m*/*z* 201^+^ and 189^+^ were due to the neutral losses of 18 Da (H_2_O) and 36 Da (2H_2_O), suggesting that Mh1 was the demethylation product of Mh0 (Figure 5B). Metabolites Mh2-Mh2a were calculated as the same formula C_15_H_16_O_5_ based on the accurate masses [M + H]^+^ at *m*/*z* 277.1070^+^, which were 2 Da (2H) less than Mh0, indicating that Mh2-Mh2b were generated from the dehydrogenation reaction. The fragmentation ions of *m*/*z* 219^+^ and 205^+^ suggested the dehydrogenation reaction occurred at the first hydroxyl group. Metabolite Mh3 was eluted at 6.12 min as C_16_H_20_O_5_ based on the accurate masses [M + H]^+^ at *m*/*z* 293.1383^+^, and it was 14 Da (CH_2_) more than Mh0. The ion at *m*/*z* 217^+^ indicated a demethylation reaction occurred at the second hydroxyl group (Figure 5C). Metabolite Mh4 was eluted at 6.05 min at *m*/*z* 295.1176^+^ and it was 16 Da (O) more than Mh0. The major fragment ions were *m*/*z* 277^+^ and 259^+^, which were 14 Da (C) and 36 Da (2H_2_O) less than *m*/*z* 295^+^, respectively. It suggested hydroxylation occurred at the terminal of the side chain (Figure 5D). Metabolites Mh5 and Mh5a exhibited the same formula (C_14_H_14_O_4_) and were eluted at 3.98 min at *m*/*z* 247.0960^+^ and 5.05 min at *m*/*z* 247.0960^+^, respectively. Mh5 and Mh5a were 32 Da (CH_2_-H_2_O) less than Mh0. It indicated demethylation and dehydration occurred. The molecular formula of Mh6 was calculated as C_14_H_14_O_5_ according to the accurate molecule [M + H]^+^ ion at *m*/*z* 263.0910^+^ and the characteristic ion at *m*/*z* 277.0697^+^, suggesting that the dehydrogenation reaction occurred. Metabolite Mh7 was confirmed as C_15_H_14_O_3_ based on the accurate molecule [M + H]^+^ ion at *m*/*z* 243.1010^+^. Compared to Mh0, it lost 36 Da (2H_2_O), which indicated dehydrations occurred. Metabolite Mh8 was eluted at 5.81 min and showed the [M + H]^+^ ion at *m*/*z* 261.1130^+^ (C_15_H_16_O_4_), which was 18 Da (H_2_O) less than Mh0.

#### 2.3.2. Structure Elucidation of ISM Metabolites

Mi0 was eluted at 8.41 min, which was calculated as C_15_H_16_O_4_ based on the protonated molecule [M + H]^+^ ion at *m*/*z* 261.1121^+^ and the fragmentation ions at *m*/*z* 243^+^, 231^+^, 189^+^, and 177^+^ described in Figure 5E. Metabolite Mi1 was calculated as C_14_H_14_O_4_ based on the [M + H]^+^ ion at *m*/*z* 247.0965^+^, and it was 14 Da (CH_2_) less than Mi0, suggesting that Mi1 was the demethylation product of Mi0. Metabolites Mi2-Mi2a were calculated as C_15_H_14_O_4_ based on the [M + H]^+^ ion at *m*/*z* 259.0965^+^ and were 2 Da (2H) less than Mi0, indicating that Mi2-Mi2a were generated from dehydrogenation, and the position of oxidation was at the side chain. Metabolites Mi3-Mi3a were deduced as C_14_H_14_O_5_ according to the accurate masses [M + H]^+^ at *m*/*z* 263.0914^+^, and they were 2 Da (O-CH_2_) more than Mi0. The fragmentation ion at *m*/*z* 245^+^ indicated a hydroxylation reaction occurred on the side chain, and the demethylation reaction occurred at the methoxy group. Metabolites Mi4-Mi4b were confirmed as C_15_H_14_O_5_ based on the [M + H]^+^ ion at *m*/*z* 275.0914^+^, indicating they were 14 Da (O-2H) more than Mi0. The fragmentation ion of *m*/*z* 247^+^ was 28 Da (CO) less than *m*/*z* 275^+^, suggesting that hydroxylation occurred at the terminal of the side chain, and the carbonyl group formed by dehydrogenation, but the position of hydroxylation was uncertain. Metabolites Mi5-Mi5c were deduced as C_14_H_14_O_5_ based on the accurate masses [M + H]^+^ at *m*/*z* 277.107^+^, which was 16 Da (O) more than Mi0, indicating that Mi5-Mi5c were hydroxylation products of Mi0, and the positions of hydroxylation occurred on the side chain. Metabolites Mi6 was confirmed as C_15_H_18_O_5_ based on the molecular ion [M + H]^+^ at *m*/*z* 279.1227^+^, and the fragmentation ion of *m*/*z* 261^+^ was formed by neutral loss of 18 Da (H_2_O). It showed that Mi6 was the hydration product of Mi0. Metabolites Mi7-Mi7c were deduced as C_15_H_16_O_6_ based on the accurate masses [M + H]^+^ at *m*/*z* 293.102^+^, which was 32 Da (2O) more than Mi0, indicating that Mi7-Mi7c were the dihydroxylation products of Mi0. In addition, the positions of dihydroxylation existed in the isopentenyl group. Metabolite Mi8 was eluted at 6.24 min and showed [M + H]^+^ at *m*/*z* 295.1176^+^, which was 34 Da (H_2_O_2_) higher than Mi0. The fragmentation ions at *m*/*z* 277^+^ and 259^+^ were due to the constant neutral losses of 18 Da (H_2_O). It indicated Mi8 was the hydrogen peroxide product of Mi0. The carbonyl group of the side chain was reduced to hydroxyl carbon, and the dihydroxylation occurred at the terminal of the side chain.

Metabolites Mi9, Mi9a, Mi10, and Mi10a were identified as adduct metabolites. Mi9 and Mi9a were deduced as C_18_H_21_NO_6_S based on the [M + H]^+^ ion at *m*/*z* 380.1162^+^. The characteristic fragmentation ions at *m*/*z* 334^+^ and 291^+^ corresponded to the neutral losses of 46 Da (COOH+2H) and 89 Da (C_3_H_7_NO_2_). These fragments were consistent with the known fragmentation pattern of cysteine, suggesting that Mi9 and Mi9a were cysteine conjugates generated from glutathione conjugate (Figure 5F). Mi10 and Mi10a exhibited the same molecular ion [M + H]^+^ at *m*/*z* 422.1268^+^, which matched the molecular formula C_20_H_23_NO_7_S. The main characteristic ion at *m*/*z* 350^+^ suggests deacetylation occurred, and cysteine conjugates were formed. The other fragmentation ions were at *m*/*z* 380^+^, 376^+^, 334^+^, and 291^+^, corresponding to the known fragmentation pattern of acetylcysteine. The neutral losses of 42 Da (CH_2_CO), 46 Da (COOH + 2H), 88 Da (COOH + 2H + CH_3_CO), and 131 Da (C_5_H_9_NO_3_) suggested that Mi10 and Mi10a were acetylcysteine conjugates generated by acetylation of the cysteine conjugate (Figure 6A).

#### 2.3.3. Structure Elucidation of MER Metabolites

Me0 was eluted at 8.15 min, which was calculated as C_15_H_16_O_4_ based on [M + H]^+^ at *m*/*z* 261.1121^+^ and the major fragmentation ions at *m*/*z* 243^+^, 231^+^, 217^+^, and 189^+^ (Figure 6B). Me1 was calculated as C_14_H_14_O_4_ based on the [M + H]^+^ ion at *m*/*z* 247.0965^+^, and it was 14 Da (CH_2_) less than Me0, suggesting that Me1 was the demethylation product of Me0. Metabolites Me2-Me2b were calculated as C_15_H_14_O_4_ based on the [M + H]^+^ at *m*/*z* 259.0965^+^, and these metabolites were 2 Da (2H) less than Me0, indicating that they were generated from dehydrogenation, and the positions of oxidation existed in the isoprenoid group. Metabolite Me3 was confirmed as C_14_H_14_O_5_ based on the [M + H]^+^ ion at *m*/*z* 263.0914^+^ and 2 Da (O-CH_2_) more than Me0. The fragmentation ions at *m*/*z* 245^+^ and 207^+^ indicate hydroxylation occurred on the benzopyran ring, but the position of substitution was uncertain. Additionally, the demethylation reaction existed on the methoxy group. Me4 was eluted at 5.43 min and showed [M + H]^+^ ion at *m*/*z* 275.0914^+^, which was 14 Da (O-2H) more than Me0. The main fragmentation ion at *m*/*z* 247^+^ was 28 Da (CO) less than *m*/*z* 275^+^, indicating hydroxylation reaction occurred at the terminal of the side chain, and the carbonyl group was formed by dehydrogenation, but the position of hydroxylation was uncertain. Metabolites Me5-Me5c were deduced as C_15_H_16_O_5_ based on [M + H]^+^ at *m*/*z* 277.107^+^, which was 16 Da (O) more than Me0, indicating that Me5-Me5c were hydroxylation products of Me0, and the positions of hydroxylation existed on the side chain. Metabolites Me6-Me6a exhibited the same formula C_15_H_18_O_5_ based on the [M + H]^+^ ion at *m*/*z* 279.1227^+^, which was 32 Da (2O) more than Me0, indicating that Me6-Me6a were hydration products of Me0, and the positions of hydration existed on the cyclic ether and pyran ring lactone, respectively. Me7 was calculated as C_15_H_14_O_6_ according to the accurate mass [M + H]^+^ ion at *m*/*z* 291.0863^+^. In addition, it neutrally lost 30 Da (2O-2H) compared with Me0. The characteristic fragmentation ion at *m*/*z* 263^+^ suggested that hydroxylation occurred and a carbonyl group formed at the end of the side chain, but the exact position of hydroxylation was uncertain. The [M + H]^+^ ion of Me8 at *m*/*z* 295.1176^+^ was 34 Da (2H + 2O) more than Me0, and the fragmentation ions at *m*/*z* 221^+^ and 203^+^ were due to the neutral losses of 18 Da (H_2_O), indicating that dihydroxylation occurred at the end of the side chain, and the carbonyl group of the side chain was reduced to hydroxyl carbon, indicating Me8 was the hydrogen peroxide product of Me0.

Metabolites Me9-Me9a, Me10, and Me11 were identified as phase II metabolites. Me9 and Me9a were deduced as C_18_H_21_NO_6_S based on the accurate mass [M + H]^+^ ion at *m*/*z* 380.1162^+^, and the characteristic fragment ions of *m*/*z* 334^+^ and 291^+^ corresponded to the neutral losses of 46 Da (COOH + 2H) and 89 Da (C_3_H_7_NO_2_), respectively. These fragments were consistent with the fragmentation pattern of cysteine, suggesting that Me9 and Me9a were cysteine conjugates from glutathione conjugate (Figure 6C). Me10 and Me10a exhibited the same molecular ion at *m*/*z* 422.1268^+^, which matched C_20_H_23_NO_7_S. The main fragment ion of *m*/*z* 350^+^ suggested deacetylation occurred and cysteine formed. Additionally, the other fragmentation ions at *m*/*z* 380^+^, 376^+^, and 334^+^ corresponded to the known fragmentation pattern of acetylcysteine and the neutral losses of 42 Da (CH_2_CO), 46 Da (COOH + 2H), and 88 Da (COOH + 2H + CH3CO), suggesting Me10 and Me10a the *N*-acetylcysteine conjugates (Figure 6D). Me11 was deduced as C_21_H_24_O_10_ based on the accurate mass [M + H]^+^ at *m*/*z* 422.1268^+^. The characteristic fragment ions of *m*/*z* 359^+^ and 261^+^ corresponded to the neutral losses of 78 Da (C_3_H_10_O_2_) and 176 Da (C_6_H_8_O_6_), respectively. These fragments were consistent with the fragmentation pattern of glucuronide, suggesting that Me11 was a glucuronide conjugation product of Me0 (Figure 6E). Me12 showed protonated molecule [M + H]^+^ ion at *m*/*z* 453.1391^+^, and the characteristic fragmentation ions of *m*/*z* 437^+^, 419^+^, and 277^+^ corresponded to the neutral losses of 16 Da (O), 34 Da (O + H_2_O), and 176 Da (C_6_H_8_O_6_) respectively. These fragments were consistent with the elimination of hydroxylation and glucuronide, suggesting that Me12 was a hydroxylation and glucuronide conjugation product of Me0 (Figure 6F).

### 2.4. CYPs Involved in the Formation of MH, ISM, and MER Metabolites

Metabolic enzymes participated in the metabolic pathways, contributing to a systematic understanding of the response of individual enzymes to drugs. To understand the metabolic pathways of C-PYCs, 13 recombinant CYP enzymes were evaluated. It was found that the dehydrogenation metabolite Mh2a was the only metabolite catalyzed by CYP1A1 (Figure 7A). Previous studies reported that MH could inhibit the enzyme activities of CYP1A2 and 2C19, and the other human CYPs showed minimal or no effect on MH metabolism [17,18]. Among the CYPs that participated in the ISM metabolism, CYP1A1 and 2B6 catalyzed the formation of demethylation metabolite Mi1 (Figure 7B). CYP1A1 was the major enzyme that catalyzed the formation of dehydrogenized metabolites Mi2a (Figure 7C). Additionally, CYP1A1 and 3A5 were the major enzymes that contributed to the formation of hydroxylation and dehydrogenation products Mi4a and Mi4b (Figure 7D,E). CYP1A1, 3A4, and 3A5 were the major enzymes responsible for the formation of hydroxylation products Mi5a-Mi5c (Figure 7F,G). As for the metabolism of MER, it showed that CYP2B6 was the major enzyme that catalyzed the formation of demethylation metabolites Me1 (Figure 7H); CYP1A1, 2C19, 3A4, and 3A5 catalyzed the formation of dehydrogenation metabolites Me2a and Me2b (Figure 7I,J). The hydroxylated metabolites Me5a and Me5b (Figure 7K) were catalyzed by CYP1A1, 2C19, 3A4, and 3A5, and the dehydroxylated metabolite Me6 was catalyzed by CYP1B1 and 2B6 (Figure 7L).

### 2.5. Anti-Inflammatory Activity of MH, ISM, and MER

#### 2.5.1. Cytotoxicity of MH, ISM, and MER in RAW 264.7 Macrophages

To evaluate the cytotoxicity of compounds MH, ISM, and MER, the viability of RAW 264.7 cells was measured by MTT assay. The result (Appendix A) showed no significant difference in cell viability between the control group and the groups treated with various concentrations of PYCs, indicating they did not affect the normal cell growth up to 200 μM (Appendix A).

#### 2.5.2. Inhibition of MH, ISM, and MER on LPS-Induced NO Production in RAW 264.7 Cells

Increased NO production is a typical inflammatory response that occurs in LPS-induced macrophages [19]. To evaluate the anti-inflammatory activity of MH, ISM, and MER, their effects on LPS-induced NO production in RAW 264.7 cells were investigated. As shown in Figure 8A, compared to the control group, NO production was significantly increased in LPS-induced RAW 264.7 cells, which was suppressed by the three C-PYCs in a dose–response manner. Interestingly, the NO inhibition of MH was slightly weaker than ISM and MER, indicating ISM and MER have better anti-inflammatory activity compared to MH, which probably correlated with the differences in the side chain in the prenyl group between the three C-PYCs.

#### 2.5.3. Inhibition of MH, ISM, and MER on LPS-Induced over Expression of Pro-Inflammatory Mediators

In inflammatory response, the production of NO is regulated by the expression of inducible NO synthase (iNOS) [20]. To determine the effects of MH, ISM, and MER on the expression of iNOS in RAW 264.7 macrophages, cells were co-treated with three coumarins and LPS. As shown in Figure 8B, the mRNA expression of iNOS was significantly upregulated in the group treated with LPS alone, and all of the three C-PYCs effectively inhibited iNOS overexpression in a dose-dependent manner. Furthermore, upregulated expression of iNOS and cyclooxygenase (COX-2) has been implicated in several chronic inflammatory diseases. To investigate their anti-inflammatory mechanism, the mRNA expression of COX-2 was also determined by quantitative reverse transcription-PCR. As shown in Figure 8C, MH, ISM, and MER effectively inhibited COX-2 mRNA overexpression induced by LPS. In addition, macrophages produce large amounts of inflammatory cytokines such as tumor necrosis factor-alpha (TNF-α), interleukin-6 (IL-6), and interleukin-1 (IL-1β) that cause chronic inflammation when an inflammatory response occurs [21]. Therefore, we further evaluated their effects on inflammatory cytokines. The results (Figure 8D–F) suggested both ISM and MER remarkably inhibited TNF-α, IL-6, and IL-1β mRNA level in a dose-dependent manner at concentrations of 200 μM and 100 μM, while MH significantly decreased TNF-α transcription only at higher concentrations up to 200 µM, indicating that MH has a weaker inhibitory effect on inflammatory cytokines transcription. However, iNOS expression is also induced in response to other inflammatory stimuli such as cytokines [22]. Thus, although MH can significantly suppress LPS-induced iNOS mRNA transcription, the release of cytokines such as IL-6 and IL-1β could also stimulate iNOS overexpression and lead to the increase in NO produce. In addition, the expression level of these inflammatory cytokines correlated with the release of NO (Appendix A). Therefore, it might be the potential reason that MH has weaker anti-inflammatory activity compared to ISM and MER.

## 3. Discussions

MH, ISM, and MER were the typical C-PYCs compounds found in Fructus aurantii, which exhibited different isoprenoid groups at the position of C2′, including hydroxyl, carbonyl, and cyclic ether groups. In this study, UPLC-ESI-QTOF-MS combined with multivariate data analysis was used to investigate the metabolic pathways of C-PYCs of Fructus aurantii. The results suggested that the lipophilicity of C-PYCs increased with the substitution of carbonyl, cyclic ether, and hydroxyl groups. In addition, the metabolic rates of MH, ISM, and MER were 8.7%, 84.1%, and 24.8% in HLM, and they were 27.1%, 81.1%, and 52.6% in MLM, respectively. This suggested that the isoprenoid group was the active site where almost all metabolic reactions occurred, demonstrating that the metabolism of the isoprenoid group was a crucial step in C-PYCs metabolism. Dehydrogenation, hydroxylation, and demethylation were the major metabolic pathways of MH. Moreover, the metabolic pathways of ISM were dehydrogenation, demethylation, hydroxylation, hydrolysis, dihydroxylation, cysteine, and *N*-acetylcysteine conjugation. Additionally, dehydrogenation, hydrolysis, hydroxylation, cysteine, ***N***-acetylcysteine, glucuronide, and hydroxylation-glucuronide conjugation were the major metabolic pathways of MER. Hydroxylation was the common pathway of C-PYCs, and the number of hydroxylation reactions, including hydroxylation, dehydrogenation, dihydroxylation, hydrogenation, increased with the substitution of hydroxyl, carbonyl, and cyclic ether groups. Hydrolysis metabolites Mi6, Me6, and Me6a were only detected in the metabolism of ISM and MER. Among them, pyran ring hydrolysis was the common pathway of ISM and MER, and cyclic ether hydrolysis was the characteristic pathway of MER. CYPs contributed to detoxification, cell metabolism, homeostasis, and drug metabolism [23,24]. The results of the in vitro metabolism of MH showed that dehydrogenized metabolite was the only product generated by CYP1A1. CYP1A1, 1A2, 2B6, 2C19, 3A4, and 3A5 were involved in the metabolic conversion of ISM. CYP1A1, 2B6, 2C19, 2D6, 3A4, and 3A5 were the primary enzymes participating in the metabolism of MER. CYP1A1 was the only common enzyme involved in the metabolism of MH, ISM, and MER, and CYP1A1, 2C19, 2B6, 3C19, 3A4, and 3A5 were the major enzymes participating in the formation of metabolites of ISM and MER. The metabolic maps of MH, ISM, and MER are provided in Figure 9 and Appendix A.

Drug metabolism plays a key role not only in the efficacy and safety of drugs but also in the discovery and development of new drugs. With first-pass metabolism elimination, drugs were converted into active or inactive metabolites, resulting in adverse therapeutic effects [25,26,27,28]. Among the three C-PYCs, MH was the most active coumarin with a low metabolic conversion rate. Therefore, it suggested that MH itself generated biological activity. ISM and MER were less active with a higher conversion rate, exhibiting anti-inflammatory, anti-mycobacterial, and anti-proliferation activity. Moreover, MER could convert into MH by hydrolysis. Although the amount of MH converted from MER was very low, it is undeniable that hydrolysis was an important metabolic pathway in terms of the bioactivity of MER. If the conversion was efficient enough, MER could exhibit stronger biological activity. Cysteine, acetylcysteine, glucuronide, and hydroxylation-glucuronide conjugates were the main phase II metabolites of C-PYCs, of which glucuronide could combine with a variety of harmful substances to exert detoxification, and cysteine could be involved in liver phospholipid metabolism and cell reduction. Therefore, we speculate that conjugation metabolites might be potential active metabolites with various biological activities.

## 4. Materials and Methods

### 4.1. Chemicals and Reagents

Meranzin hydrate (MH), isomeranzin (ISM). and meranzin (MER) were purchased from BioBioPharm (Kunming, China). Mouse liver microsomes (MLMs) and human liver microsomes (HLMs) were purchased from Bioreclamayionivt Inc. (Hickville, NY, USA). CYPs were purchased from Xenotech LLC (Kansas City, KS, USA). NADPH, Lipopolysaccharide (LPS), dimethyl sulfoxide (DMSO), and Griess reagent were purchased from Sigma-Aldrich CO. (St. Louis, MO, USA). 5-Diphenyl-2*H*-tetrazolium bromide (MTT) and dexamethasone (DEX) were purchased from Solarbio (Beijing, China). Trizol reagent was purchased from Invitrogen (Carlsbad, CA, USA). Dulbecco’s modified Eagle’s medium (DMEM), trypsin-EDTA, and penicillin–streptomycin solution were purchased from Hyclone (Logan, UT, USA). Fetal bovine serum (FBS) was purchased from ExCell Bio (Shanghai, China). HiScript III RT SuperMix and SYBR Green PCR Master Mix were purchased from Vazyme (Nanjing, China). Other chromatographic-grade and analytical-grade reagents were purchased from Thermo Fisher (Waltham, MA, USA).

### 4.2. In Vivo Metabolism of MH, ISM, and MER

In this study, 6–8-week-old C57BL/6J male mice (20–22 g) were purchased from Hunan Slac Jingda Laboratory Animal Co., Ltd. (Hunan, China). Mice were fasted for 12 h before administration and had free access to water in temperature- and humidity-controlled conditions with a 12 h dark/light cycle. Mice were randomly divided into four groups (*N* = 4 each group), including control group, MH treatment group, ISM treatment group, and MER treatment group. The control group was orally administrated with corn oil alone, and the other groups were orally administrated with three C-PYCs by gavage to mice at a dose of 35 mg/kg. The mice were euthanized 24 h after administration. Samples of plasma, urine, and feces were collected. All animal experiments were carried out in accordance with the Institute of Laboratory Animal Resources guidelines and approved by the Institutional Animal Care and Use Committee of West China Hospital, Sichuan University (No. 20211266A).

### 4.3. In Vitro Metabolism of Coumarins

The in vitro incubation for microsome metabolism was carried out as described previously [14]. The experiment was carried out in phosphate-buffered saline (PBS) solution, and 180 µL of the incubation system contained drug (50 μM), HLM (0.5 mg/mL), MLM, or 2 pmol/mL of each cDNA-expressed CYP (control, 1A1, 1A2, 1B1, 2A6, 2B6, 2C19, 2C8, 2C9, 2D6, 2E1, 3A4, 3A5, and 4A11) in 96-well plates. After pre-incubation at 37 °C for 5 min with shaking at 800 rpm, 10 mM NADPH (20 µL) was added to the incubation system, and the absence of drug and NADPH were positive and negative controls, respectively. After incubation at 37 °C for 40 min with shaking at 800 rpm, 200 µL of ice-cold acetonitrile was added to terminate the reaction and eliminate the microsome protein. A 5 µL aliquot of the supernatant was injected into the UPLC-ESI-QTOFMS system for sample analysis after centrifugation at 18,000× *g* for 20 min at 4 °C. The MLM, HLM, and CYP incubation experiments were conducted in triplicate.

### 4.4. Sample Preparation

Mice were placed in metabolism cages after treatment. Urine and feces samples were collected from dose to 24 h. Blood samples were collected from mice orbit at 1, 3, and 24 h post-dosing, and the plasma samples were obtained by centrifugation at 2000× *g* for 5 min at 4 °C. All samples were stored at −80 °C until analysis. The preparation of samples (feces, urine, and plasma) was processed as reported previously [29]. Briefly, chlorpropamide (5 µM) in acetonitrile was used as the internal standard in this study. Feces samples were extracted by adding tenfold 50% acetonitrile and shocked for 20 min at room temperature. Next, samples were centrifuged at 18,000× *g* for 20 min to precipitate protein. The supernatant (100 µL) was transferred to a new centrifuge tube and diluted with 200 µL acetonitrile. After centrifugation at 18,000× *g* for 20 min, 5 µL of supernatant was injected into the UPLC-ESI-Q-TOFMS system for analysis. For the urine and plasma samples, 20 µL of urine was mixed with 180 µL of 50% acetonitrile, and 10 µL of plasma sample was mixed with 190 µL of 67% acetonitrile. After urine and plasma samples were centrifuged (18,000× *g*, 20 min), 5 µL of supernatant was subjected to the UPLC-ESI-QTOF-MS system for analysis.

### 4.5. UPLC-ESI-QTOFMS Analysis

The urine, feces, and microsome samples were analyzed by the UPLC-ESI-QTOF/MS system (Agilent, Santa Clara, CA, USA). Metabolites showed good separation in the Agilent 1290 infinity UPLC system (Agilent Technologies, Santa Clara, CA, USA) equipped with an XDB-C18 column (2.1 mm × 100 mm, 1.8 µM). The column temperature was maintained at 45 °C, and the flow was set at 0.3 mL/min. Elution was performed using gradient elution ranging from 2% to 98% acetonitrile, containing 0.1% formic acid for 16 min. The injection volume was 5 µL, and the mass signals of ions were collected in both positive (ESI +) and negative (ESI −) modes with electrospray ionization. Nitrogen was used as the collision gas and drying gas, which was set at 350 °C and 9 L/min. Nebulizer pressure was set at 35 psi, and the capillary voltage was set at 3.5 kV. The structures of metabolites were identified by the accurate mass measurements compared to the fragmentary mode of the parent compound, and the MS/MS chromatogram of metabolites was obtained using four collision energy, 10, 15, 20, and 30 eV. The MS was calibrated using the ESI-L Low-Concentration Tuning Mix (Agilent, Santa Clara, CA, USA).

### 4.6. Multivariate Data Analysis

The mass signals were obtained by the MassHunter WorkStation data acquisition software (Agilent, Santa Clara, CA, USA), and the raw mass spectrum data were processed by Mass Profinder and Mass Profiler Professional software (Agilent, Santa Clara, CA, USA). The accurate *m/z*, retention time, and peak area were gained from the multivariate data matrix. Principal component analysis (PCA) and orthogonal projection to latent structures-discriminant analysis (OPLS-DA) were used to identify the major latent variables and the potential metabolites by SIMCA-P + 13.0 software. Qualitative Analysis of MassHunter Acquisition Data software (Agilent, Santa Clara, CA, USA) was used to identify the metabolites. Moreover, the hydrogenation and sodium peaks were used to screen metabolites in the positive mode (ESI +), and the condition of the extracted ion chromatogram (EIC) was set at ± 20 ppm. The relative abundance was evaluated based on the peak areas of ions and normalized by the peak area of internal standard, and the sum of peak areas of total detected ion counts was integrated as 100% in in vivo and in vitro analyses. Experimental values were presented as mean ± SD by Prism v. 6 (GraphPad Software, San Diego, CA, USA). The raw data were normalized and scaled in Simca-P software using the Par mode.

### 4.7. Anti-Inflammatory Assays

#### 4.7.1. Cell Culture

Immortalized mouse myoblast cell line RAW 264.7 was obtained from the Cell and Molecular Biology Public Laboratory of West China Hospital. Cells were grown in DMEM and supplemented with 10% fetal bovine serum (FBS), 100 unit/mL penicillin, and 100 µg/mL streptomycin. The cells were incubated at 37 °C in a humidified atmosphere containing 5% CO_2_ [19].

#### 4.7.2. Cell Viability Assay

RAW 264.7 macrophages were seeded in 96-well plates at a density of 10^5^ cells in each well [19,30]. After overnight culture, MH, ISM, and MER were applied to the cells for 24 h. DMSO was used as a control, and the concentration of DMSO treated in the cells did not exceed 0.5% (*v*/*v*). Then, 20 µL of MTT solution (1 mg/mL) was added to the culture supernatant and incubated for 2 h. Cell culture supernatant was then removed, and 200 µL of DMSO was added to each well [19]. The absorbance (OD) was measured at 570 nm using the BioTek Epoch2 microplate spectrophotometer.

#### 4.7.3. NO Inhibition Assay

RAW 264.7 cells were plated into 96-well plates at a density of 1 × 10^5^ in each well and cultured overnight. Cells were pre-treated with MH, ISM, MER (200 µM, 100 µM), and the positive control dexamethasone (DEX, 100 µM) for 1 h [30,31], followed by co-treatment with LPS (1 µg/mL) for another 24 h at 37 °C. After the treatment, the cell culture supernatant was collected and mixed with standard Griess reagent. Nitrite, a stable metabolite of NO in aqueous solution, was measured by the absorbance (OD) at 550 nm [19,30,32].

#### 4.7.4. Real-Time PCR

RAW 264.7 cells were seeded in 12-well plates at a density of 1 × 10^6^ and cultured overnight. Cells were pre-treated for 1 h with MH, ISM, and MER at concentrations of 200 µM and 100 µM. DEX (100 µM) was used as the positive control. Then, LPS (1 µg/mL) was added to stimulate inflammation for 24 h. Total RNA was isolated from cell pellets with TRIzol reagent, reverse transcribed to cDNA, and subjected to quantitative PCR [11,19]. The program for amplification was 1 cycle of 95 °C for 30 s, followed by 40 cycles of 95 °C for 10 s, 55 °C for 30 s, and 72 °C for 40 s. Relative gene expression levels were normalized to GAPDH expression levels. Sequences of primers used for PCR amplification are shown below:

(1) iNOS: GTTCTCAGCCCAACAATACAAGA (Forward), GTGGACGGGTCGATGTCAC (Reverse); (2) Cox2: TGACCCCCAAGGCTCAAATAT (Forward); TGAACCCAGGTCCTCGCTA (Reverse); (3) TNFα: CCACCACGCTCTTCTGTCTAC (Forward), AGGGTCTGGGCCATAGAACT (Reverse); (4) IL6: CGGAGAGGAGACTTCACAGAGGA (Forward), TTTCCACGATTTCCCAGAGAACA (Reverse); (5) IL1β: CCCTGCAGCTGGAGAGTGTGGA (Forward), TGTGCTCTGCTTGTGAGGTGCTG (Reverse); (6) GAPDH: TTGATGGCAACAATCTCCAC (Forward), CGTCCCGTAGACAAAATGGT (Reverse).

### 4.8. Statistical Analysis

Data were reported as means ± SD of three independent tests. All experiments were repeated three times. Unpaired Student’s *t*-test was used to identify significant differences between means. Statistical analysis was carried out with GraphPad Prism 4 (GraphPad Software, La Jolla, CA, USA). In all cases, *p* < 0.05 was assumed to indicate significant differences.

## 5. Conclusions

In the present study, metabolic maps of MH, ISM, and MER were determined in vivo and in vitro by the UPLC-ESI-Q-TOFMS system, contributing to the understanding of the characterizations of C-PYCs. The major metabolic pathways of C-PYCs included hydroxylation, dehydrogenation, demethylation and conjugation. CYP1A1, 2B6, 3A4, and 3A5 were the major metabolic enzymes for the formation of C-PYCs metabolites. The isoprenoid groups of C-PYCs played an important role not only in the metabolism but also in the biological activity of C-PYCs. The anti-inflammatory effect of ISM and MER was stronger than MH, potentially because hydroxylation and conjugation were their primary metabolic pathways. The findings could provide a basis for the further investigation of C-PYC function in vivo.

## Figures and Tables

**Figure 1 molecules-26-06558-f001:**
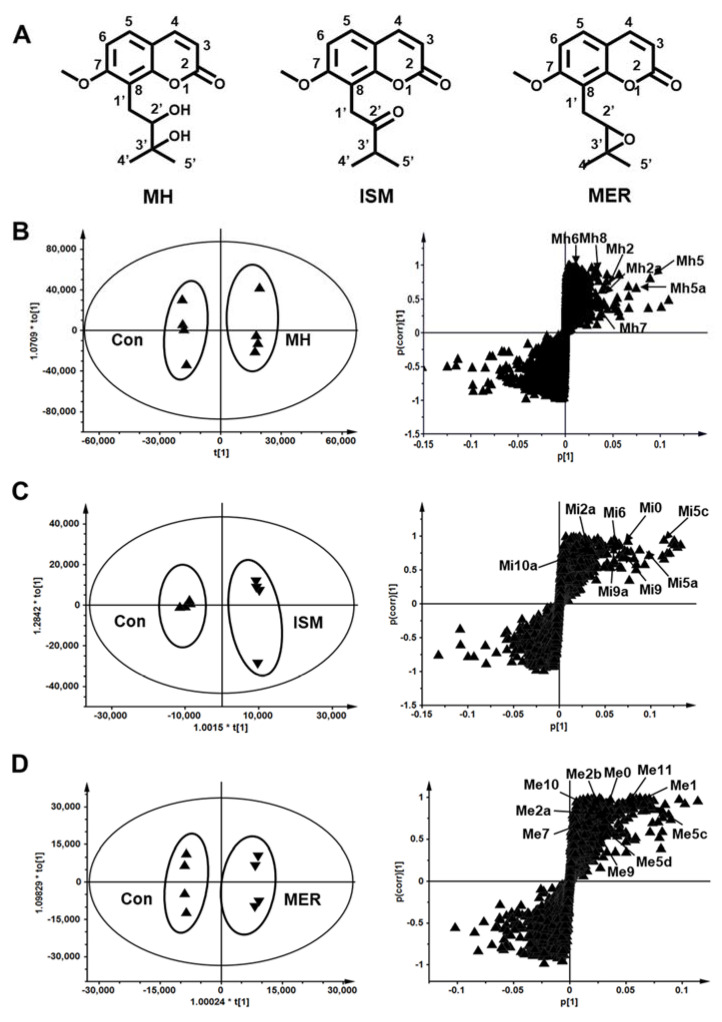
Structure of coumarins and metabolic profiling of MH, ISM, and MER in mice. (**A**) Meranzin hydrate (MH), isomeranzin (ISM), meranzin (MER). (**B**) Scores plot of OPLS-DA and S-plot analysis from control and MH-treated mice urine. (**C**) Scores plot of OPLS-DA and S-plot analysis from control and ISM-treated mice urine. (**D**) Scores plot of OPLS-DA and S-plot analysis from control and MER-treated mice urine.

**Figure 2 molecules-26-06558-f002:**
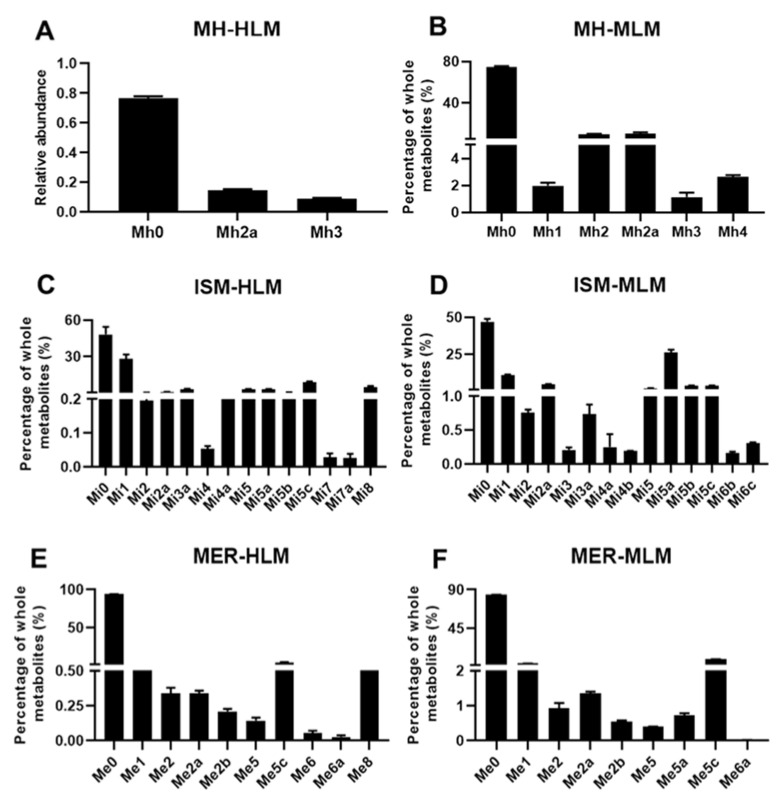
In vitro metabolism of MH, ISM, and MER. Relative abundance of MH metabolites in human liver microsomal (HLM) (**A**) and mouse liver microsomal (MLM) incubation systems (**B**). Relative abundance of ISM metabolites in human liver microsomal (HLM) (**C**) and mouse liver microsomal (MLM) incubation systems (**D**). Metabolites of MER in human liver microsomal (HLM) (**E**) and mouse liver microsomal (MLM) incubation systems (**F**).

**Figure 3 molecules-26-06558-f003:**
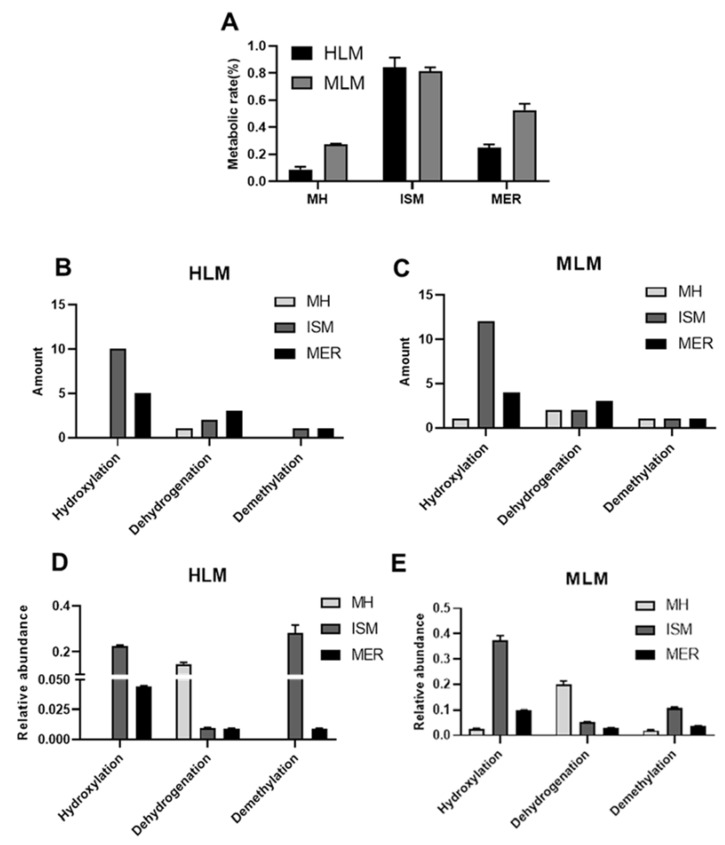
Metabolism of MH, ISM, and MER in liver microsome. (**A**) Metabolic rate of MH, ISM, and MER in HLM and MLM. Amount of metabolic pathways of MH, ISM, and MER in HLM (**B**) and MLM incubation systems (**C**). Relative abundance of metabolic pathways of MH, ISM, and MER in HLM (**D**) and MLM incubation systems (**E**).

**Figure 4 molecules-26-06558-f004:**
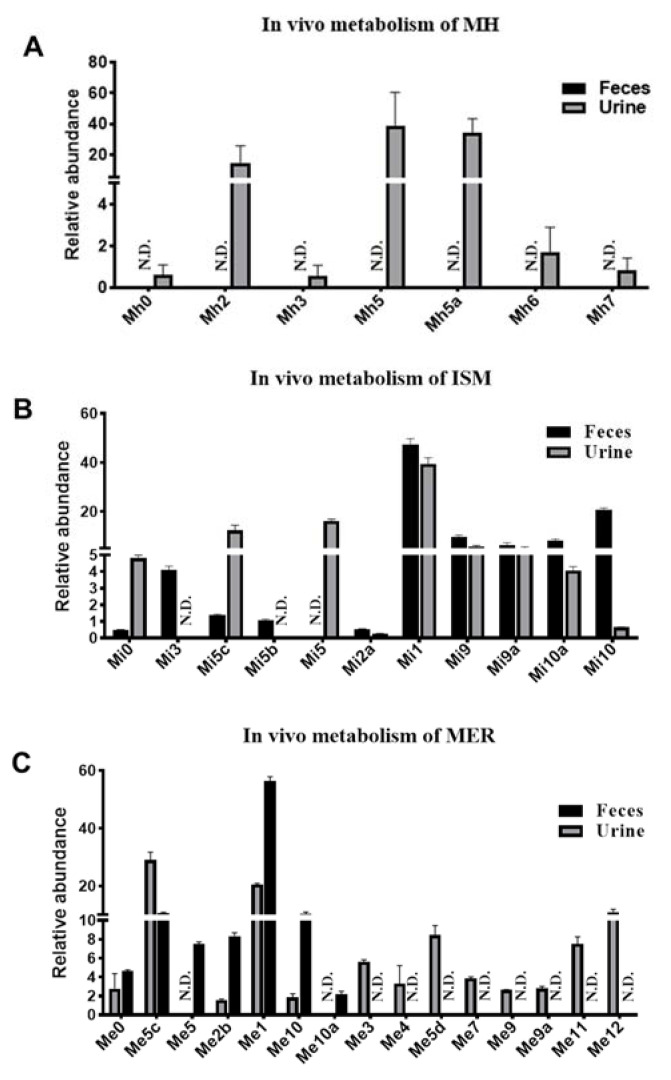
Metabolomics analysis of MH, ISM, and MER metabolites in mouse feces and urine. (**A**) Comparison of MH major metabolites in mouse feces and urine. (**B**) Comparison of ISM major metabolites in mouse feces and urine. (**C**) Comparison of MER major metabolites in mouse feces and urine.

**Figure 5 molecules-26-06558-f005:**
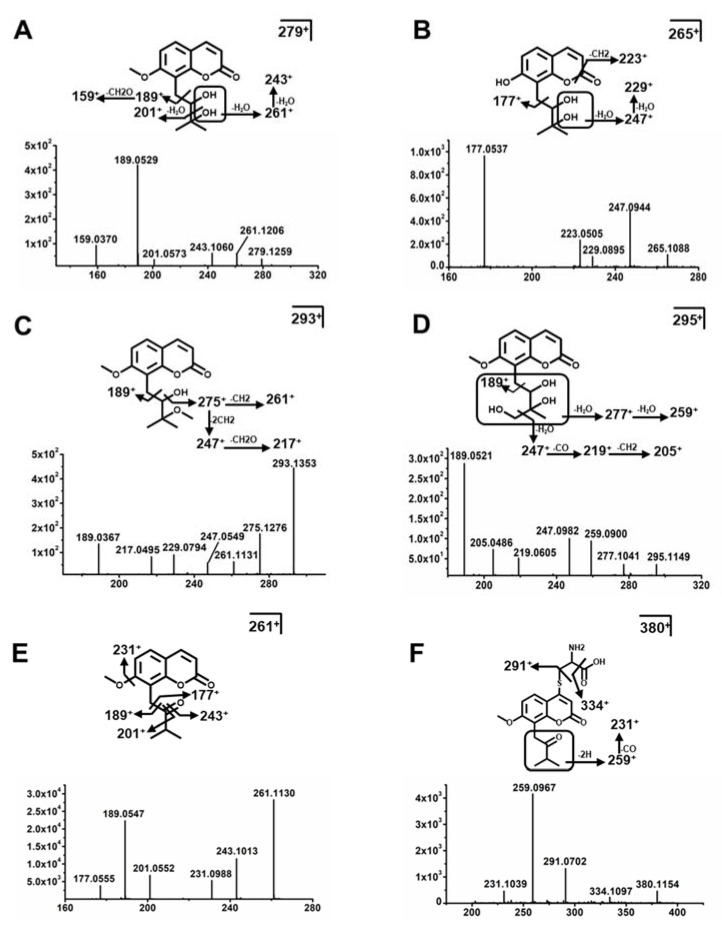
Typical MS/MS fragments of MH and ISM metabolites. MS/MS fragment and structures of Mh0 (**A**), Mh1 (**B**), Mh3 (**C**), and Mh4 (**D**). MS/MS fragment and structures of Mi0 (**E**), Mi9/Mi9a (**F**).

**Figure 6 molecules-26-06558-f006:**
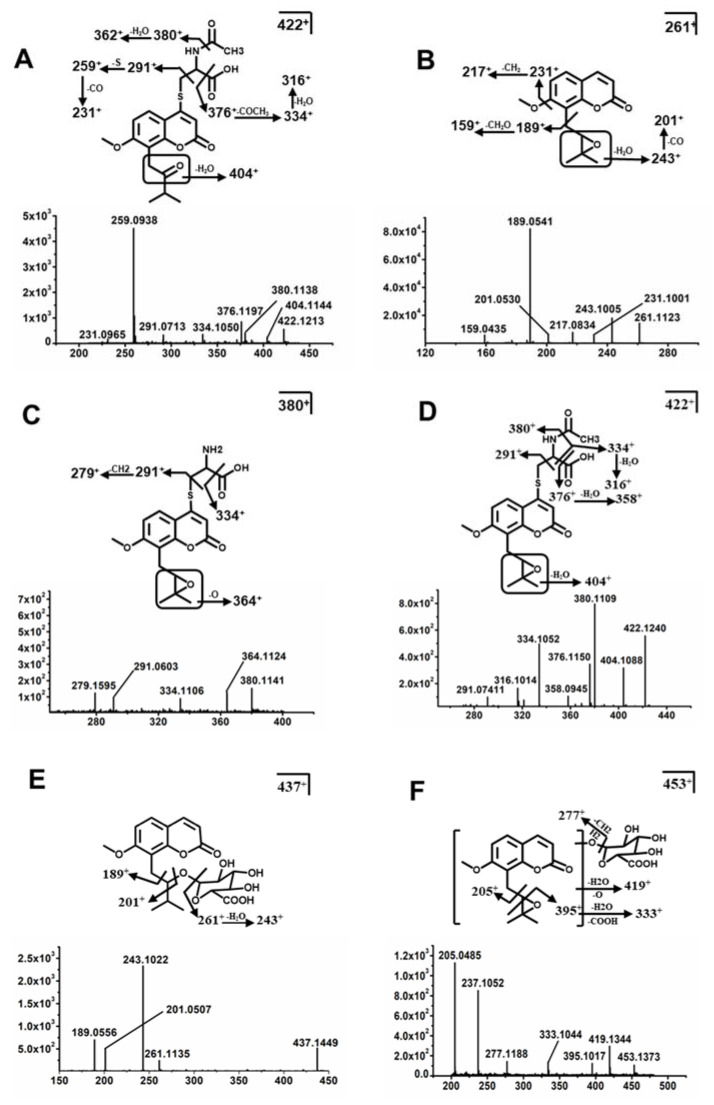
Typical MS/MS fragments of ISM and MER metabolites. MS/MS fragment and structures of Mi10/Mi10a (**A**). MS/MS fragment and structures of Me0 (**B**), Me9/Me9a (**C**). Me10/Me10a (**D**), Me11 (**E**), and Me12 (**F**).

**Figure 7 molecules-26-06558-f007:**
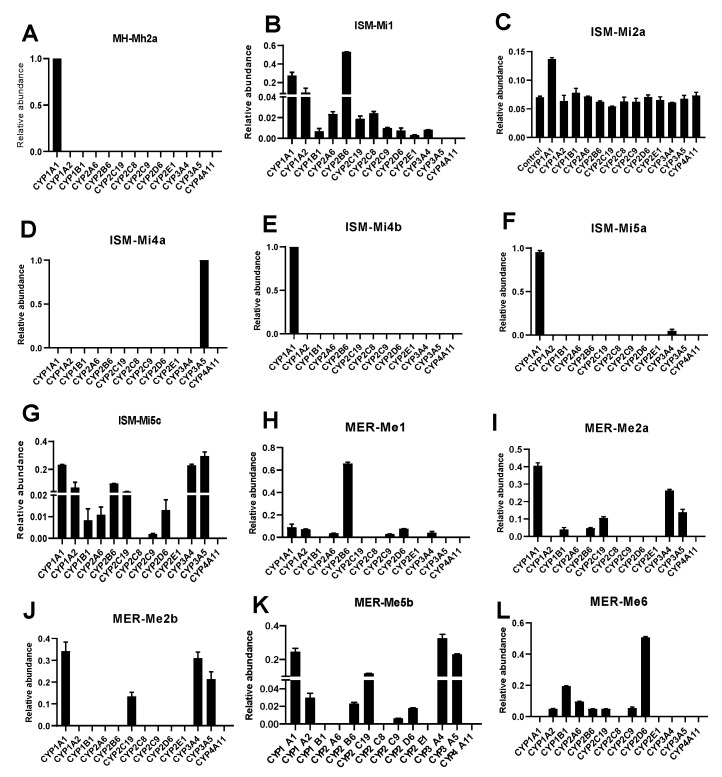
Contribution of CYPs to the formation of in vitro metabolites of MH, ISM, and MER. Relative abundance of CYPs contribute to formation of Mh2a (**A**), Mi1 (**B**), Mi2a (**C**), Mi4a (**D**), Mi4b (**E**), Mi5a (**F**), Mi5c (**G**), Me1 (**H**), Me2a (**I**), Me2b (**J**). Me5c (**K**), and Me6 (**L**).

**Figure 8 molecules-26-06558-f008:**
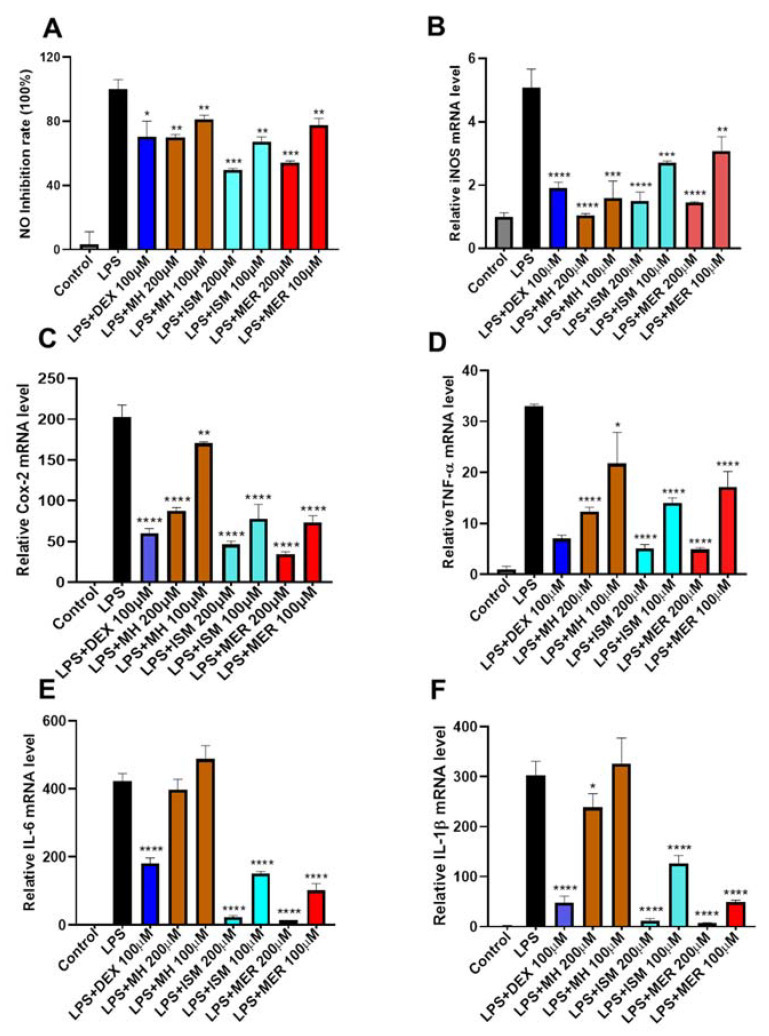
Anti-inflammatory effect of MH, ISM, and MER in RAW 264.7 cells. Data represented the mean ± SEM of three independent experiments in triplicate. * *p* < 0.05, ** *p* < 0.01, *** *p* < 0.005, **** *p* < 0.001 vs. LPS group. (**A**) Inhibition of MH, ISM, and MER on LPS-induced NO production. (**B**) Inhibition of MH, ISM, and MER on LPS-induced iNOS mRNA level. (**C**) Inhibition of MH, ISM, and MER on LPS-induced Cox-2 mRNA level. (**D**) Inhibition of MH, ISM, and MER on LPS-induced TNF-α mRNA level. (**E**) Inhibition of MH, ISM, and MER on LPS-induced IL-6 mRNA level. (**F**) Inhibition of MH, ISM, and MER on LPS-induced IL-1β mRNA level.

**Figure 9 molecules-26-06558-f009:**
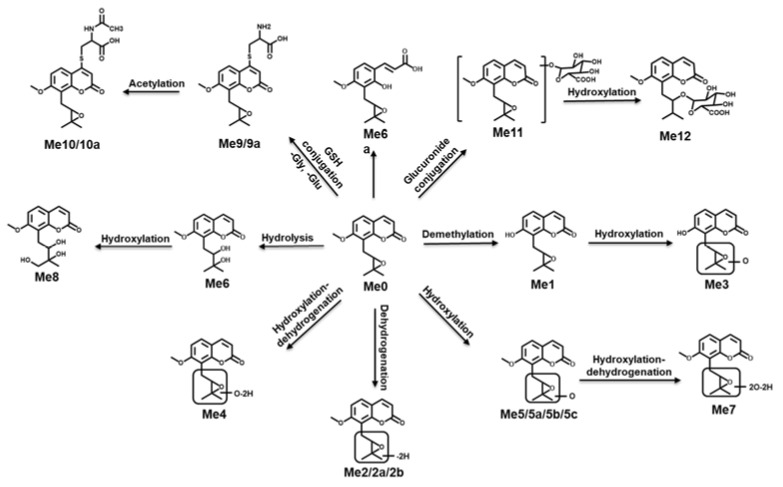
Metabolic map of MER.

**Table 1 molecules-26-06558-t001:** Summary of MER metabolites produced in vivo and in vitro metabolism.

Metabolite	Molecular Formula	RT (min)	*m*/*z* [M + H]^+^	Mass Error (ppm)	MS/MS Fragments	Identification	Source
Me0	C_15_H_16_O_4_	8.15	261.1121	0.27	243, 231, 217, 189	Meranzin	P, F, U, H, M
Me1 *	C_14_H_14_O_4_	6.79	247.0965	−1.13	229, 221, 201, 187	Mer-CH_2_	U, F, H, M
Me2 *	C_15_H_14_O_4_	6.47	259.0965	3.94	243, 231, 217, 201, 189	Mer-2H	H, M
Me2a *	C_15_H_14_O_4_	6.85	259.0965	−8.03	243, 231, 217, 189	Mer-2H	H, M
Me2b *	C_15_H_14_O_4_	7.68	259.0965	−6.49	243, 229, 217, 203, 189	Mer-2H	F, U, H, M
Me3 *	C_14_H_14_O_5_	6.67	263.0914	4.20	245, 207, 191, 175	Mer + O-CH_2_	U
Me4 *	C_15_H_14_O_5_	5.43	275.0914	5.83	259, 247, 231, 219, 205	Mer + CH_2_	U
Me5 *	C_15_H_16_O_5_	6.47	277.1070	−3.05	259, 247, 219, 205	Mer + O	F, H, M
Me5a *	C_15_H_16_O_5_	6.83	277.1070	7.78	259, 243, 227, 217, 205	Mer + O	M
Me5b *	C_15_H_16_O_5_	7.67	277.1070	4.90	259, 233, 217, 205	Mer + O	F, U, H, M
Me5c *	C_15_H_16_O_5_	7.84	277.1070	−7.02	261, 243, 215, 205, 191	Mer + O	U
Me6 *	C_15_H_18_O_5_	6.29	279.1227	9.70	261, 243, 217, 207, 189	Mer + H_2_O	H
Me6a *	C_15_H_18_O_5_	7.70	279.1227	−7.14	261, 235, 221, 207, 189	Mer + H_2_O	H, M
Me7 *	C_15_H_14_O_6_	6.68	291.0863	0.65	275, 263, 245, 217, 203	Mer + O + CH_2_	U
Me8 *	C_15_H_18_O_6_	5.62	295.1176	7.42	259, 231, 221, 203, 189	Mer + 2O + 2H	H
Me9 *	C_18_H_21_NO_5_S	5.36	380.1162	1.24	334, 305	Mer + Cysteine	U
Me9a *	C_18_H_21_NO_5_S	5.84	380.1162	3.08	364, 334, 291	Mer + Cysteine	U
Me10 *	C_20_H_23_NO_7_S	7.11	422.1268	0.02	404, 380, 376, 362, 334	Mer + *N*-acetylcysteine	F, U
Me10a *	C_20_H_23_NO_7_S	7.35	422.1268	7.02	404, 380, 358, 376, 334	Mer + *N*-acetylcysteine	F
Me11 *	C_21_H_24_O_10_	6.12	437.1442	4.08	359, 261, 243, 189	Mer + Gluc	U
Me12 *	C_21_H_24_O_11_	4.95	453.1391	2.36	437, 419, 395, 291, 237	Mer + O + Gluc	U

a/b/c isomer metabolite; *, undescribed metabolite. P, plasma; F, feces; U, urine; H, HLM; M, MLM.

## Data Availability

Not applicable.

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
