# Peer review of "Metabolic Profile of C-Prenyl Coumarins Using Mass Spectrometry-Based Metabolomics"

_molecules, 2021, doi:10.3390/molecules26216558_

Round 1
Reviewer 1 Report
This section 2.2 explaining in vitro metabolism experiments should be at first and numered 2.1 and after this section with profiling explanation( In vitro metabolism of MH, ISM, and MER by HLM and MLM)
4.5. UPLC-ESI-QTOFMS analysis (in this section put information how MS was calibrated)
4.6. Multivariate data analysis (data were normalization and scalling or not ?)
Line 90 MLM and HLM (add explanation in this position)
Line 477 should be nm not nM
When authors used qualitative analysis ,were supported by Library or literature to identified of particular metabolome ?
Author Response
Response to Reviewer 1 Comments
Point 1: This section 2.2 explaining in vitro metabolism experiments should be at first and numered 2.1 and after this section with profiling explanation ( In vitro metabolism of MH, ISM, and MER by HLM and MLM)
Response 1: The order of section 2.2 and 2.1 has been switched. And the number of figures in two sections was updated.
Point 2: 4.5. UPLC-ESI-QTOFMS analysis (in this section put information how MS was calibrated)
Response 2: The MS was calibrated using ESI-L low concentration tunning mix, which was provided in the section of “4.5. UPLC-ESI-QTOFMS analysis”.
Point 3: 4.6. Multivariate data analysis (data were normalization and scalling or not?)
Response 3: The raw data were normalized and scaled in Simca-P software using Par mode, which was provided in the section of “4.6. Multivariate data analysis”.
Point 4: Line 90 MLM and HLM (add explanation in this position)
Response 4: The full name of MLM and HLM has been provided.
Point 5: Line 477 should be nm not nM
Response 5: Revised.
Point 6: When authors used qualitative analysis, were supported by Library or literature to identified of particular metabolome?
Response 6: The metabolites identified by metabolomics belong to xenobiotic metabolite. The structures of metabolites were identified by the accurate mass measurements and comparing to the fragmentary mode of parent compound. 52 metabolites were identified and all of them were found to be novel metabolites. The library and literature were not available for the identification. These information was provided in the section of “4.5. UPLC-ESI-QTOFMS analysis”.
Reviewer 2 Report
The author should change the name of the plant says Fructus Aurantia and should say Fructus aurantia, the species is written with lowercase letter.
Abbreviations whose meaning is not indicated are used, for example, LPS (line 27) review.
In line 167 Mi0 is in bold, it must be changed
The citation should be reviewed, they are not written according to "instruction to the authors"
Author Response
Response to Reviewer 2 Comments
Point 1: The author should change the name of the plant says Fructus Aurantia and should say Fructus aurantia, the species is written with lowercase letter.
Response 1: Fructus Aurantia has been changed to Fructus aurantii.
Point 2: Abbreviations whose meaning is not indicated are used, for example, LPS (line 27) review.
Response 2: The full name of lipopolysaccharide (LPS) was provided in Line 27.
Point 3: In line 167 Mi0 is in bold, it must be changed
Response 3: Changed.
Point 4: The citation should be reviewed, they are not written according to "instruction to the authors"
Response 4: The format of literature was revised according to "instruction to the authors".